# Ultrasound Biomicroscopy as a Novel, Potential Modality to Evaluate Anterior Segment Ophthalmic Structures during Spaceflight: An Analysis of Current Technology

**DOI:** 10.3390/diagnostics14060639

**Published:** 2024-03-18

**Authors:** Benjamin Soares, Joshua Ong, Daniela Osteicoechea, Cihan Mehmet Kadipasaoglu, Ethan Waisberg, Prithul Sarker, Nasif Zaman, Alireza Tavakkoli, Gianmarco Vizzeri, Andrew G. Lee

**Affiliations:** 1Department of Medicine, Boston University Chobanian & Avedisian School of Medicine, Boston, MA 02118, USA; 2Department of Ophthalmology and Visual Sciences, University of Michigan Kellogg Eye Center, Ann Arbor, MI 48105, USA; 3Texas A&M School of Medicine, Bryan, TX 77807, USA; 4Department of Ophthalmology, Blanton Eye Institute, Houston Methodist Hospital, Houston, TX 77094, USA; 5Department of Ophthalmology, University of Cambridge, Cambridge CB0QQ, UK; 6Human-Machine Perception Laboratory, Department of Computer Science and Engineering, University of Nevada Reno, Reno, NV 89557, USA; 7Department of Ophthalmology, University of Texas Medical Branch, Galveston, TX 77555, USA; 8Center for Space Medicine, Baylor College of Medicine, Houston, TX 77030, USA; 9The Houston Methodist Research Institute, Houston Methodist Hospital, Houston, TX 77094, USA; 10Departments of Ophthalmology, Neurology, and Neurosurgery, Weill Cornell Medicine, New York, NY 10065, USA; 11Department of Ophthalmology, University of Texas MD Anderson Cancer Center, Houston, TX 77030, USA; 12Department of Ophthalmology, The University of Iowa Hospitals and Clinics, Iowa City, IA 52242, USA

**Keywords:** spaceflight associated neuro-ocular syndrome (SANS), ultrasound biomicroscopy (UBM), space medicine

## Abstract

Ocular health is currently a major concern for astronauts on current and future long-duration spaceflight missions. Spaceflight-associated neuro-ocular syndrome (SANS) is a collection of ophthalmic and neurologic findings that is one potential physiologic barrier to interplanetary spaceflight. Since its initial report in 2011, our understanding of SANS has advanced considerably, with a primary focus on posterior ocular imaging including fundus photography and optical coherence tomography. However, there may be changes to the anterior segment that have not been identified. Additional concerns to ocular health in space include corneal damage and radiation-induced cataract formation. Given these concerns, precision anterior segment imaging of the eye would be a valuable addition to future long-duration spaceflights. The purpose of this paper is to review ultrasound biomicroscopy (UBM) and its potential as a noninvasive, efficient imaging modality for spaceflight. The analysis of UBM for spaceflight is not well defined in the literature, and such technology may help to provide further insights into the overall anatomical changes in the eye in microgravity.

## 1. Introduction

With plans for future long-duration crewed voyages to the Moon and Mars on the horizon, NASA has identified several potential health risks to astronauts in space. One such risk is spaceflight-associated neuro-ocular syndrome (SANS), a collection of neurologic and ophthalmic findings including optic disc edema (ODE), globe flattening, retinal nerve fiber layer thickening, chorioretinal folds, hyperopic shifts, and cotton-wool spots (CWS) [1,2,3]. SANS has been documented both during and after spaceflight. A postflight questionnaire given to 300 individuals indicated that 29% of short (space shuttle) and 60% of long-duration mission flyers (ISS) experienced a degradation in distant and near visual acuity [2].

Despite continued efforts, the complete etiology of SANS remains unclear [4]. A microgravity-induced cephalad fluid shift and presumed secondary cervical and cerebral venous congestion with impairment of CSF outflow is thought to be a main contributor. Additional contributors including lymphatic stasis, genetic, inflammatory, and metabolic features may also influence progression [2].

SANS imaging has utilized modalities such as fundoscopy, optical coherence tomography (OCT), magnetic resonance imaging (MRI), and orbital/cranial ultrasound. Intraflight imaging platforms aboard the ISS currently include fundoscopy, OCT, and orbital/cranial ultrasound [2]. Imaging has been critical in our understanding of SANS, as manifestations of disease pathology have often been subclinical. Continued advancements in our capabilities to temporally monitor SANS pathology in microgravity environments will undoubtedly allow for better understanding of this complex disease.

Research on SANS has focused on the neuro-ophthalmic changes that occur in the posterior segment and optic nerve. This is reflected in imaging modalities that offer excellent resolution of the retina in OCT and fundoscopy [2,3]. The anterior segment can be visualized in cross-section by orbital ultrasound. Orbital ultrasound is a cost-effective technique that can visualize macroscopic ocular structures. Ultrasound was first used to image ocular structures by Mundt and Hughes (A-Scan) and Baum and Greenwood (B-Scan) in the 1950s. Transducer frequency in standard orbital ultrasound is typically around 10 MHz. While orbital ultrasound can grossly visualize the eye, more precision is required to image anterior segment structures in detail. To study potentially subtle, microgravity-induced changes, a higher-resolution technique is necessary.

Ultrasound biomicroscopy (UBM) was introduced as a noninvasive ophthalmic imaging technique in 1990 by Foster and Pavlin, with the ability to produce micrometer-resolution cross-sectional images of the anterior segment [5]. While traditional ophthalmic ultrasound uses a transducer with a 10 MHz frequency, UBM uses frequency ranges between 50 and 100 MHz. This results in higher resolution images, with resolving powers up to 20 um axially and 50 um laterally [6].

Much of the anterior segment can be identified on cross-sectional UBM images, including the cornea, iris, anterior chamber, scleral spur, Schlemm’s canal, ciliary body, lens capsule, and anterior lens. UBM has mainly been used clinically to assess angle closure, corneal, and lens pathologies [7,8]. Along with clinical changes associated with SANS, other risk factors to astronauts in long-duration spaceflight are corneal damage and cataract formation. UBM, therefore, has multiple potential use cases for ocular imaging on future missions.

Another modality with potential for use in high-resolution imaging of the anterior segment is anterior segment optical coherence tomography (AS-OCT). AS-OCT can provide measurements of anterior chamber depth, anterior chamber width, anterior chamber volume, and measurements involving the scleral spur [9,10]. Advantages of AS-OCT compared to UBM include ease of use, as AS-OCT does not require an expert operator. However, UBM has improved depth of resolution when compared to AS-OCT, with the ability to image structures behind the iris including the ciliary body and lens. While AS-OCT may have significant potential as an anterior segment imaging modality in spaceflight, this review focuses on UBM.

## 2. Materials and Methods

PubMed Database and Google Scholar were queried with keywords used independently or in conjunction, including “Spaceflight Associated Neuro-Ocular Syndrome”; “SANS”; “Ultrasound Biomicroscopy”; “UBM”; “Ultrasound”; and “Microgravity”. A total of 85 papers were reviewed and 83 were included in this paper. Papers were excluded that did not include relevant information regarding UBM metrics, or focused on posterior segment applications of UBM.

## 3. Current Clinical Application of UBM

UBM can provide high-resolution visualization and quantitative assessment of anterior segment structures (Figure 1). In a study analyzing 95 eyes from 52 adults, anterior segment depth (ASD), anterior chamber angle (ACA), and thickness of the iris and the ciliary body were measured. Mean ASD was 2.92 mm, and mean ACA was 34.3 degrees. Iris thickness was measured in three places, the root of the iris, the middle of the iris, and the juxtapupillary edge of the iris, with mean measurements of 0.41, 0.51, and 0.71 mm in each location, respectively [11]. A second study investigating iris thickness found mean thickness at the root of the iris to be 0.41, mean thickness at the middle of the iris to be 0.51, and mean thickness at the juxtapupillary margin to be 0.71 mm [12].

Three major categories of ocular pathology in which UBM has been established as a clinical tool for assessment include angle closure, corneal, and lens pathologies.

In angle closure, apposition of the iris and trabecular meshwork occludes aqueous outflow and can cause a dangerous elevation in intraocular pressure. Subsequent damage to the optic nerve is associated with visual field loss. Angle closure can be divided into primary and secondary etiologies. Primary angle closure is mainly caused by pupillary block, where the iris occludes aqueous humor outflow. Secondary angle closure is caused by an identifiable pathology. Multiple anatomic sites can instigate angle closure through a variety of mechanisms, including the iris (pupillary block angle closure), the ciliary body (plateau iris angle closure), the lens (phacomorphic glaucoma), and the posterior chamber (malignant glaucoma) [6]. Risk factors for angle closure include hyperopia, family history of angle closure, advancing age, female gender, Asian or Inuit descent, and thicker lens [13]. Two additional risk factors of particular importance, because of their potential alteration in microgravity, are shorter axial length and shallow anterior chamber depth.

Choosing the effective treatment paradigm is reliant on the initial cause of angle closure; therefore, UBM can provide valuable diagnostic information given its high-resolution imaging of relevant structures [14,15]. High agreement was found between the gold standard in angle closure assessment, gonioscopy, and UBM when both were performed in a dark room [16].

**Figure 1 diagnostics-14-00639-f001:**
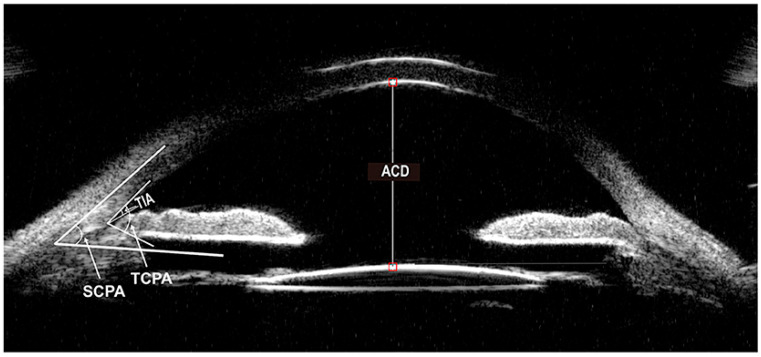
Ultrasound biomicroscopy (UBM) of the anterior segment of the eye showcasing various metrics including anterior chamber length (ACD), trabecular-meshwork ciliary process angle (TCPA), scleral ciliary process angle (SCPA), and trabecular iris angle (TIA). These anterior segment metrics are quantifiable to monitor during spaceflight. Reprinted with permission from [15] under Creative Commons License (https://creativecommons.org/licenses/by/4.0/legalcode.en (accessed on 1 February 2024)).

UBM can be used for qualitative analysis, such as confirmation of angle closure, existence of ciliary rotation, or the identification of other abnormalities of the angle [17]. Efforts have been made to quantify angle closure via UBM. Angle opening distance (AOD) and trabecular-iris angle (TIA) are two such measurements [5,18]. AOD is the most commonly measured parameter in the assessment of angle closure. To obtain an AOD measurement, the scleral spur is first identified, and then a point on the internal wall of the corneoscleral plane at a given distance from the scleral spur (often either 250 or 500 microns) is located. From this point, a line perpendicular to the plane of the trabecular surface is drawn to meet the surface of the iris. This line is the AOD, and is referred to as AOD250 or AOD500 based on how far from the scleral spur the first point was. The average at AOD500 for healthy eyes was found to be 347 ± 181 microns [17,18]. The TIA is a measurement of the angle formed by two lines, one passing through a point 500 microns from the scleral spur, and a second passing through a point perpendicularly opposite the first one on the iris.

These single parameter measurements are relatively inconsistent between individuals because of variations in iris curvature. A refined parameter was proposed that accounted for variations in anatomy by measuring an area, called the angle recess area (ARA), which considers an the area enclosed by multiple measurements [18,19]. 

UBM has demonstrated value in the assessment of corneal pathologies. This tool was found to be an accurate and reproducible method for measuring corneal thickness, and differentiation of the cross-sectional structure of the cornea is possible [17,20,21]. Accurate measurements of corneal thickness can be made. UBM was found to be superior to computed tomography and/or B-scan ultrasound in the detection of corneal ocular foreign bodies [22]. In the case of anterior surface tumors, UBM can be used to assess the depth and layer of origin. This allows for effective and accurate treatment planning, especially surgical interventions [23]. Corneal hydrops, a condition characterized by movement of aqueous humor through a small tear in Descemet’s membrane, can also be assessed with UBM. Corneal edema and detachment of Descemet’s membrane are well visualized, and progression can be monitored [24]. Additionally, UBM has been utilized to assess keratoconus and post-surgical outcomes of corneal transplantation [25,26,27].

Additionally, UBM has demonstrated value in the assessment of lenticular pathologies [28]. Nuclear and cortical cataracts can be visualized on UBM as regions of elevated internal reflectivity [29]. In the setting of cataract surgery, UBM can be used to assess post-surgical outcomes. Postoperative complications can be closely monitored following congenital cataract surgery, especially for those with media opacities or when pupil dilation is not possible [28]. A common complication of cataract surgery is retention of nuclear fragments. UBM can detect retained nuclear fragments posterior to the iris plane postoperatively [29,30]. Another use of UBM is in the placement and monitoring of intraocular lenses (IOLs). Preoperative high-resolution evaluation of the ocular anatomy can be useful to estimate postoperative IOL position. With phakic refractive lens implants, UBM can specifically guide the surgeon in determination of sulcus plan diameter at specific meridians, avoiding future problems with over or undersized implants [31].

## 4. Logistics of Ultrasound Biomicroscopy in Spaceflight

The use of ultrasound in spaceflight dates to the 1970s, when NASA and Russian space programs began testing ultrasound imaging in spaceflight for research purposes. Eight different imagers were tested between the 1970s and the mid-1990s [32,33]. Ultrasound use in spaceflight typically requires two astronauts, an examiner and examinee. NASA’s Advanced Diagnostic Ultrasound in Microgravity (ADUM) study indicated that remote operation of imaging equipment in space was possible with terrestrial guidance [33,34]. Over 82 percent of surveyed students of this study agreed or strongly agreed that their educational experience with ultrasound equipment was positive. Although UBM examination is a more complex and intricate procedure, these results are reassuring, especially given that novice UBM examiners demonstrated a high degree of reproducibility using the Paradigm Model P45 UBM Plus [35].

Before an ultrasound session, ISS astronauts are required to set up hardware including a laptop, ultrasound keyboard, monitor, and probes. Each scan lasts between 20 and 50 min, with scanning sessions lasting for 2 h total. Examinees must be physically restrained with elastic cords or fabric belts to ensure positional stability in microgravity.

One major difference between orbital A/B-Scan ultrasound and UBM is the requirement for a coupling gel. The high-frequency UBM transducer necessitates a coupling gel to be placed on the surface of the eye to guarantee signal transmission. During a UBM examination, topical anesthesia is applied to the ocular surface, before a specially designed eyecup (22–24 mm diameter) is used to separate the eyelids and form a water bath environment. This environment is filled with a viscous, sonolucent coupling fluid such as methylcellulose (1–2.5%) [17].

The patient is examined in a supine position facing the ceiling. After topical anesthesia, a specially designed eyecup (22–24 mm diameter) is used to separate the eyelids and form a water bath environment. This is filled with a viscous, sonolucent coupling fluid such as methylcellulose (1–2.5%). Some examiners use normal saline to fill the cup after sealing the interface between the eye and the base of the cup with 2.5% methylcellulose. In the microgravity environment of space, acoustic coupling has been achieved with water droplets in spaceflight, potentially removing the need for a coupling gel such as methylcellulose [36]. Targeted experimentation is warranted to evaluate the viability of water as an acoustic coupler with UBM.

## 5. UBM for Assessment of Pressure-Related Changes to the Eye

A transient increase in IOP upon entry to microgravity has been observed [2,3]. During the first 15 min of microgravity exposure, IOP, as measured by tonometry, had increased 92% compared to baseline [37]. Proposed mechanisms for this initial increase in IOP include a cephalad fluid redistribution, choroidal expansion, and increased episcleral venous pressure [1,2,3,38,39]. SANS imaging has focused on posterior segment changes, with orbital/cranial ultrasound being the only visualization of the anterior segment obtained so far. More precise and high-resolution imaging of the anterior segment could reveal microgravity-induced changes.

Secondary angle closure resulting in the acute elevation of IOP can present as an ocular emergency. In addition to increased IOP, a risk factor for angle closure is shallow anterior chamber depth (ACD) [40]. Shallow ACD is associated with both increased risk of angle closure and increasing age, and is regarded as the cardinal risk factor for angle closure [41,42]. While orbital ultrasound and optical biometry can produce measurements of the anterior chamber, micrometer-resolution measurements made by UBM would be more valuable in the assessment of future angle closure risk.

Recent investigations beginning to probe the relationship of the anterior segment in SANS have revealed a decreased anterior chamber depth in astronauts who participated in long-duration space flight. The ACD of astronauts on board the ISS for a 6-month mission was measured pre- and postflight via optical biometry (IOLMaster 500; Carl Zeiss Meditec, Jena, Germany [43]. Upon immediate return from spaceflight, ACD had decreased by a mean of 3%. Decrease in ACD remained below preflight values for up to 1 year after return to Earth. The authors highlight the need for intraflight measurement of ACD and other ocular structural changes. Imaging via UBM could be critical in documenting the temporal profile of these changes which may increase the risk of angle closure and consequent loss of vision. Additionally, preflight UBM could identify those with decreased angle recess area, narrow iridocorneal angle, and anterior chamber depth to prevent future spaceflight-induced acceleration of angle closure.

Following the elevation in IOP that occurs in astronauts upon initial exposure to microgravity, a return to baseline of IOP has been documented [2,3,37]. A compensatory decrease in aqueous volume enabled by alterations in the conventional outflow pathway is thought to contribute [2]. The mechanism of this compensatory decrease is unclear and mysterious, especially given the sustained cephalad fluid shift, as evidenced by an elevation in jugular venous pressure that remains throughout long-duration spaceflight missions [2,44]. The conventional outflow pathway, accounting for approximately 75–90% of aqueous outflow, involves the movement of aqueous humor from the anterior chamber into the trabecular meshwork and then Schlemm’s canal before exiting the eye via episcleral veins [45,46]. According to the simplified Goldmann equation, factors affecting IOP include aqueous humor production, facility of trabecular outflow, and episcleral venous pressure [47]. Histologic studies have indicated a strong correlation between outflow capacity and dimensions of outflow pathway sites [48]. With UBM, significant decreases in the coronal diameter of SC and thickness of SC have been visualized in patients with primary open angle glaucoma [8]. In addition, conversely, an increased diameter of SC and thickness of SC may contribute to increased aqueous humor egress from the eye, allowing IOP to return to baseline and compensating for the pressure-induced cephalad fluid shift caused by microgravity. Intraflight use of UBM to visualize components of the conventional outflow pathway could reveal the mechanism for the normalization of IOP during prolonged exposure to microgravity. Importantly, an investigation of this compensatory mechanism could contribute to understanding of physiologic regulation of IOP, and glaucoma research as a whole.

While IOP does not appear to be chronically elevated during spaceflight, some of the main findings of SANS, including posterior globe flattening, choroidal folds, and ODE, reflect ocular structural changes. This could be related to the translaminar pressure gradient that forms between intracranial and intraocular compartments [49]. While anatomic changes in the anterior segment have not been grossly observed on MRI or cranial/orbital ultrasound, high-resolution imaging is warranted to investigate potential subtle changes that could negatively affect vision [2,3].

One potential countermeasure that has been suggested targets the translaminar pressure gradient. Through the use of positive-pressure goggles, intraocular pressure can be artificially elevated to reduce the translaminar pressure gradient and potentially alleviate symptoms of SANS [50]. UBM could be a valuable tool in monitoring ocular structures in response to this artificial elevation in intraocular pressure, as any evidence of angle closure or alterations in anterior chamber structures would warrant immediate termination.

## 6. Cornea Risks during Spaceflight

A comprehensive review of ocular trauma and conditions occurring during NASA spaceflight missions (including space shuttle and ISS) documented 70 corneal abrasions, 4 dry eyes, 4 eye debris, 5 complaints of ocular irritation, 6 chemical burns, and 5 ocular infections [51]. Ocular trauma is a major risk factor to astronauts from a variety of sources.

Human exposure to lunar dust has been investigated since the Apollo program. After exploration of the lunar surface, the spacesuits of astronauts were coated in large amounts of dust that was then carried into the interior of the spacecraft. Following liftoff from the moon and entrance into microgravity, this lunar dust became airborne and was reportedly irritating to the eyes of astronauts. At the time, this problem was solved by wearing their helmets until spacecraft air filters could clear the cabin of dust particles [52,53]. The problem of airborne dust in microgravity has been suggested to increase the opportunity for ocular exposure and injury [51,52,54,55,56].

Controlled experimentation on lunar dust as an ocular irritant was performed both in vitro and in vivo. Pulverized lunar dust was exposed to a human-derived epidermal keratinocyte model cultured from stratified corneal epithelium. Additionally, an in vivo model using rabbit eyes was examined. Following exposure, gross observation of treated eyes and fluorescein staining with an ocular transilluminator revealed minimal irritation [52].

Given that increased construction work is expected during the establishment of new orbital and lunar habitats, and exposure will be long term on the order of months to years, further investigation is warranted. Additionally, higher-resolution imaging of the corneal surface could suggest subtle damage that was unable to be noticed on gross observation. It has been shown that chronic insult of lunar dust in as low quantities as 20 mg/m^3^ elicits a molecular response in corneal tissue [51,55]. UBM could be used to assess microscopic damage caused by minute dust particles.

Additional corneal injuries that could occur in space include chemical and thermal injuries in the setting of prolonged CO^2^ and heat exposure, corneal edema, corneal infections, and radiation damage to the cornea [51,57]. An investigation of the effect of radiation dose on ocular complications revealed significantly higher incidence of dry eye and corneal irritation for eyes receiving higher doses of direct radiation [58]. As radiation is routinely used therapeutically in the treatment of head and neck cancers, corneal complications of direct radiation have been observed. A 44-year-old-man developed a corneal epithelial abnormality associated with conjunctival and corneal inflammation after radiation therapy for maxillary cancer. He experienced pain, loss of vision, and eventual conjunctival epithelialization of the upper and lower cornea [59].

## 7. Cataract Risks during Spaceflight

The negative effects of radiation on ocular tissue, in particular the crystallin components of the lens, have been documented [60,61]. Cataracts have been investigated as a consequence of the increased radiation exposure experienced by astronauts in space. The NASA Study of Cataract in Astronauts (NASCA) was a longitudinal study that studied the severity and progression of nuclear, cortical, and posterior subcapsular lens (PSC) opacities [62]. Subject groups included astronauts who had flown at least one mission in space, astronauts who had not yet flown in space, military aircrew personnel, and a control group. Cataracts were imaged with a Nidek EAS 1000 anterior segment camera system. Spaceflight-exposed astronauts demonstrated an increase amount and variability in cortical cataracts when compared to non-spaceflight-exposed astronauts and controls when age was controlled for, results that have been correlated with another study [63]. Another study confirmed that astronauts’ cataracts were most commonly cortical in location, also observing that United States Air Force and Navy aviator’s cataracts were mostly located in the PSC region [64].

Of note, this study determined that there are increased cataract risks at smaller radiation doses than previously reported. An additional study confirmed that relatively low doses of space radiation are linked to increased incidence and early appearance of cataracts [65].

As NASA standards limit exposure for American astronauts to 600 mSv over a career while other international space agencies allow for up to 1000 mSv, and a 1–2 yearlong Mars mission is likely to exceed this limit [51,60], research into long-term effects of low-dose radiation exposure are warranted. UBM can be used to monitor cataract formation or progression, given its sensitivity to changes in lens opacity [66].

## 8. Intraocular Lenses in Spaceflight

The safety of intraocular lenses (IOLs) in terrestrial aviation has been established, and their use is currently approved by all three US military services and the Federal Aviation Administration [67,68,69,70].

IOLs appear to be safe for short-duration spaceflight, as a 64-year-old NASA astronaut with bilateral IOLs demonstrated stable vision during a 2-week flight. IOL position was unchanged before and after the mission, and subjective visual assessment was excellent in all phases of flight [67,71].

Furthermore, IOLs were found to be well tolerated during long-duration spaceflight as well. An astronaut with unilateral acrylic IOLs implanted after cataract phacoemulsification completed a 6-month mission with excellent subjective visual assessment and no documented change in IOL position [72].

Despite these findings, IOLs are currently disqualifying for astronaut selection [67,73]. As the astronaut population increases in age, and with long-duration moon and Mars missions in the future, IOLs will undoubtedly be present in spaceflight environments. UBM can be used to analyze lens position and IOL placement [6,31]. Newer tools such as 3D-UBM provide additional visualization by showing the exact location of IOLs, insertion points of IOL haptics, and the condition of the surrounding tissues [74]. This could prove useful in examining potential subtle changes to IOL position induced by microgravity or radiation damage.

## 9. Future Directions

There is currently limited literature and research on imaging of the anterior segment during microgravity. Our research group is analyzing the biometrics of UBM to further understand the effects of microgravity on the eye. Prior to any deployment in spaceflight such as the International Space Station (ISS), our first goal will be to analyze this imaging modality in head-down tilt bed rest (HDTBR). HDTBR is a terrestrial analog for SANS [75]. While there is no perfect analog for SANS on Earth, the goal is to mimic the cephalad fluid shifts in microgravity and analyze the changes. In addition to gaining further insights into anterior segment changes, these changes can be compared to UBM in true microgravity to further understand HDTBR as an analog to ocular changes in spaceflight. We also plan to merge this imaging modality with functional testing with head-mounted technology and artificial intelligence tools for SANS to continue understanding the visual changes in microgravity [76,77,78,79,80,81]. This research may be supplemented with parabolic flight to mimic a weightlessness environment [82,83].

## 10. Conclusions

UBM is a noninvasive modality that can produce high-resolution cross-sectional images of ocular structures. Its use as a clinical tool includes the assessment of angle closure glaucoma, and corneal and lens pathologies. With long-duration spaceflight missions, maintaining ocular health in the austere environment of outer space is a challenging objective. Corneal trauma, cataract formation, and angle closure are specific use cases for UBM during spaceflight.

## Data Availability

No new data were created or analyzed in this study. Data sharing is not applicable to this article.

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
