# Peer review of "Ultrasound Biomicroscopy as a Novel, Potential Modality to Evaluate Anterior Segment Ophthalmic Structures during Spaceflight: An Analysis of Current Technology"

_diagnostics, 2024, doi:10.3390/diagnostics14060639_

Round 1

Reviewer 1 Report

Comments and Suggestions for Authors

The title of the article may confuse the readers. Reading the proposed title, it seems that the authors will write about specific data collected during the spaceflight. In truth, authors just reflect about the possible application of UBM. Maybe the title should be modified to better represent the topic of the article.
Moreover, why did the authors choose UBM and not anterior segment OCT (AS-OCT)? UBM, as all the echographic examination, are linked to the operator, whereas the AS-OCT is more standardized. Please include some information about AS-OCT in the introduction section and justify the choice; please include the following references:

1.      Changes in Anterior Segment Morphology and Intraocular Pressure after Cataract Surgery in Non-glaucomatous Eyes. Klin Monbl Augenheilkd. 2023 Apr;240(4):449-455. English. doi: 10.1055/a-2013-2374. 

2.      Anterior segment-optical coherence tomography and diabetic retinopathy: Could it be an early biomarker? Photodiagnosis Photodyn Ther. 2022 Sep;39:102995. doi: 10.1016/j.pdpdt.2022.102995.

Materials and methods section: what about exclusion criteria? When did the author perform their research on databases? The methods are not properly described to reproduce the research.

Current Clinical application of UBM section: Why did the authors write so much about clinical application of UBM? It seems not strictly necessary. This section should be brief and include quantitative data about anterior segment structure. Please, add the following references:

1.      Anterior Ocular Biometrics as Measured by Ultrasound Biomicroscopy. Healthcare (Basel). 2022 Jun 24;10(7):1188. doi: 10.3390/healthcare10071188. PMID: 35885715

2.      Ultrasound Biomicroscopy Measurements of the Normal Thickness for the Ciliary Body and the Iris in a Middle East Population. Clin Ophthalmol. 2022 Jan 10;16:101-109. doi: 10.2147/OPTH.S297977.

Logistics of ultrasound biomicroscopy in spaceflight section: 243-246: sentence is repeated. The author have already reported this information in lines 238-242. Moreover, this description seems not essential in the text because itis a well described and standardized procedure.

UBM for assessment of pressure-related changes to the eye section: There are no data reported about UBM and changes of anterior chamber, neither about changes related to increase of IOP

Cornea risks during spaceflight section: The authors proposed UBM to assess microscopic corneal damages. Why did they not consider AS-OCT?

The most interesting section of the work is the “future directions section”.
Some transcription mistakes are presented in the text.  

Comments on the Quality of English Language

Several sentences in the text are not clear. There are typing errors.

Author Response

Dear Reviewer #1

Thank you for your time and consideration in reviewing our manuscript. We have revised the manuscript according to your suggestions as follows. 

Comment 1: The title of the article may confuse the readers. Reading the proposed title, it seems that the authors will write about specific data collected during the spaceflight. In truth, authors just reflect about the possible application of UBM. Maybe the title should be modified to better represent the topic of the article.

Response to Comment 1: Thank you for the comment. We’ve decided to update the title to “Review of Ultrasound Biomicroscopy: A Novel Modality with Potential to Evaluate Anterior Segment Ophthalmic Structures During Spaceflight.

Comment 2: Moreover, why did the authors choose UBM and not anterior segment OCT (AS-OCT)? UBM, as all the echographic examination, are linked to the operator, whereas the AS-OCT is more standardized. Please include some information about AS-OCT in the introduction section and justify the choice; please include the following references:

Response to Comment 2: Thank you for the comment. We agree with the reviewer that AS-OCT may be another viable tool for anterior segment imaging during spaceflight. From lines 120-129, we introduce the capabilities of AS-OCT. We highlight its superiority in remote operations, and explain that one advantage of UBM is improved depth of resolution. We also appreciate the reviewer’s suggestion of references, as they illustrate the quantitative capabilities of AS-OCT. In the future, we hope to expound upon AS-OCT as a counterpart to UBM, as this tool certainly has potential for use in spaceflight ophthalmic imaging.

Comment 3: Materials and methods section: what about exclusion criteria? When did the author perform their research on databases? The methods are not properly described to reproduce the research.

Response to Comment 3: Thank you for the comment. We updated the materials and methods section to include exclusion criteria in lines 136-138, and two papers that were not included after initial review due to their focus on posterior segment applications of UBM rather than anterior segment applications.

Comment 4: Why did the authors write so much about clinical application of UBM? It seems not strictly necessary. This section should be brief and include quantitative data about anterior segment structures.

Response to Comment 4: Thank you for the comment. We agree with the reviewer that quantitative data is essential in understanding the use case of UBM in spaceflight, and we added a paragraph from lines 140-150 discussing some important metrics measured by UBM. We greatly appreciate the reviewer providing these references, as the quantitative data will certainly improve readers understanding of the value of UBM as a clinical tool.

Comment 5: Logistics of ultrasound biomicroscopy in spaceflight section: 243-246: sentence is repeated. The author have already reported this information in lines 238-242. Moreover, this description seems not essential in the text because itis a well described and standardized procedure.

Response to Comment 5: Thank you for the comment. We agree with the reviewer and have omitted lines 243-245.

Comment 6: UBM for assessment of pressure-related changes to the eye section: There are no data reported about UBM and changes of anterior chamber, neither about changes related to increase of IOP.

Response to Comment 6: Thank you for the comment. We agree that data regarding UBM and changes of the anterior chamber is essential in pressure related changes to the eye. We believe UBM has great potential in imaging changes in the outflow apparatus of aqueous humor in in the trabecular meshwork and Schlemm’s canal regions. From lines 331-337, we highlight previous uses of UBM as a tool to image these structural changes terrestrially, which we believe would have relevance in spaceflight as well.

Comment 7: The authors proposed UBM to assess microscopic corneal damages. Why did they not consider AS-OCT?

Response to Comment 7: Thank you for the comment. We agree that AS-OCT is another tool that could provide high-resolution imaging of corneal damage. AS-OCT, due to it’s noninvasive technique, may even be better than UBM for corneal imaging. In future works, we hope to investigate AS-OCT as a counterpart to UBM for spaceflight ophthalmic imaging. 

Reviewer 2 Report

Comments and Suggestions for Authors

This is a complitly new subject and might be of interest in next decads. However the paper dose not give any new facts other than that UBM might be use in space flights.

The paper dose give a well & detailed review of probable eye problems that may occur in space flights.

Author Response

Dear Reviewer #2,

Thank you for your time and consideration in reviewing our manuscript. 

Reviewer 3 Report

Comments and Suggestions for Authors

In the present review the authors discuss the potential use of biomicroscopy as a future method for evaluating eye health during spaceflight. The authors also review common eye issues associated with spaceflight beyond just SANS, which is a high priority topic for NASA at the moment and therefore is a current bias in contemporary literature. The review is generally well written with a few awkward sentences here and there. I was pleased to see the logistics section in the review as this is where the real clinical/diagnostic value of the review is. Similarly, the clear use of testing with HDT bedrest is a valuable explanation of part of the path to flight certifying a new diagnostic. Lastly, the discussion of lenses is valuable in light of the growing diversity of astronauts/spaceflight participants with the burgeoning commercialization of human spaceflight.

Minor points-

Abstract "barriers" should probably be barrier

Line 81 might at "features" after "metabolic" if so might change "Additional factors" on line 80 to "Additional contributors"

Line 243 might comment on how UBM would be conducted on ISS/other space platform. Specifically, would you recommend examination in a supine position facing away from Earth so as to maximize the gravity force similar to how it is done on Earth? Conversely, would you recommend a different gravity vector (recall g on ISS is roughly 90% of what it is on Earth).

Comments on the Quality of English Language

Two main issues I think require editing were highlighted in the minor comments. There are some other sentences I found awkward but if the authors think they have clearly communicated and I understood then I do not feel other changes are required vs. discretionary.

Author Response

Dear Reviewer #3,

Thank you for your time and consideration in reviewing our manuscript. We have made revisions according to your suggestions as follows. 

Comment 1: Abstract "barriers" should probably be barrier

Response to comment 1: Thank you for your comment, we have modified the abstract in line 49.

Comment 2: Line 81 might at "features" after "metabolic" if so might change "Additional factors" on line 80 to "Additional contributors"

Response to comment 2: Thank you for your comment, we have modified the grammar on lines 80-81.

Comment 3: Line 243 might comment on how UBM would be conducted on ISS/other space platform. Specifically, would you recommend examination in a supine position facing away from Earth so as to maximize the gravity force similar to how it is done on Earth? Conversely, would you recommend a different gravity vector (recall g on ISS is roughly 90% of what it is on Earth).

Response to comment 3: Thank you for your comment. We are intrigued by the reviewer’s suggestion involving astronaut position during spaceflight imaging and gravity vector selection. In future works, we hope to embellish upon the current review by constructing a more detailed protocol for UBM imaging in spaceflight.

Round 2

Reviewer 1 Report

Comments and Suggestions for Authors

The title is still not attractive according to my point of view.  I suggest the following one: "Ultrasound Biomicroscopy as a Novel Modality with Potential to Evaluate Anterior Segment Ophthalmic Structures During Spaceflight:a narrative review."

Comments on the Quality of English Language

I only suggest a minor editing of the English at this point.

Author Response

Dear Reviewer 1,

Thank you very much for your comment and your reviews that have allowed for optimization of our manuscript. It is highly appreciated. We have highly considered it and how to make the title more fitting; we would like to integrate your comment in the change to our title of "Ultrasound Biomicroscopy as a Novel, Potential Modality to Evaluate Anterior Segment Ophthalmic Structures During Spaceflight: An Analysis of Current Technology.